# Validation and Adaptation of the Multidimensional Prognostic Index in an Older Australian Cohort

**DOI:** 10.3390/jcm8111820

**Published:** 2019-11-01

**Authors:** Kimberley Bryant, Michael J. Sorich, Richard J. Woodman, Arduino A. Mangoni

**Affiliations:** 1Discipline of Clinical Pharmacology, College of Medicine and Public Health, Flinders University and Flinders Medical Centre, Bedford Park, 5042, South Australia, Australia; michael.sorich@flinders.edu.au; 2Flinders Centre for Epidemiology and Biostatistics, College of Medicine and Public Health, Flinders University, Bedford Park, 5042, South Australia, Australia; richard.woodman@flinders.edu.au

**Keywords:** Multidimensional Prognostic Index, adverse outcomes, validation, predictive tools

## Abstract

Background and aims: The Multidimensional Prognostic Index (MPI), an objective and quantifiable tool based on the Comprehensive Geriatric Assessment, has been shown to predict adverse outcomes in European cohorts. We conducted a validation study of the original MPI, and of adapted versions that accounted for the use of specific drugs and cultural diversity in the assessment of cognition, in older Australians. Methods: The capacity of the MPI to predict 12-month mortality was assessed in 697 patients (median age: 80 years; interquartile range: 72–86) admitted to a metropolitan teaching hospital between September 2015 and February 2017. Results: In simple logistic regression analysis, the MPI was associated with 12-month mortality (Low risk: OR reference group; moderate risk: OR 2.50, 95% CI: 1.67–3.75; high risk: OR 4.24, 95% CI: 2.28–7.88). The area under the receiver operating characteristic curve (AUC) for the unadjusted MPI was 0.61 (0.57–0.65) and 0.64 (95% CI: 0.59–0.68) with age and sex adjusted. The adapted versions of the MPI did not significantly change the AUC of the original MPI. Conclusion: The original and adapted MPI were strongly associated with 12-month mortality in an Australian cohort. However, the discriminatory performance was lower than that reported in European studies.

## 1. Introduction

The global population is continuously expanding and ageing as a result of significant advances in human health and favourable economic conditions [1]. However, this phenomenon also imposes significant public health and financial challenges [2]. For example, older adults often suffer from the unwanted consequences of acute and chronic diseases, particularly reduced quality of life, disability and loss of independence [3]. The introduction of the Comprehensive Geriatric Assessment (CGA) into geriatric care and the establishment of dedicated hospital wards and staff has improved the standard of care for older patients [4]. At the same time, hospitalists are ever more expected to accelerate patient turn-over, reduce the length of hospital stay and readmission rate, and arrange appropriate post-discharge follow-up. However, accurately predicting and identifying the most suitable care pathways for older in-patients, with multiple co-morbidities and significant inter-individual variability in organ function and homeostatic capacity, is problematic. 

The addition of screening tools being implemented alongside the CGA has furthermore contributed to improved care [5,6,7,8]. This has led to the development of clinical prediction rules that combine previously established tools for predicting adverse outcomes in older adults. One of them, the Multidimensional Prognostic Index (MPI), is an objective and quantifiable CGA-based tool, based on eight domains, that is used to estimate short- and medium-term adverse outcomes in older patients [9]. 

A number of studies have validated the MPI in different sub-populations suffering from acute and chronic conditions either in hospital or in the community [10,11,12,13,14,15,16,17,18,19]. However, there are some potential limitations of the MPI. Firstly, the MPI has not been investigated in non-European populations by investigators not involved in the original studies. The latter have primarily been conducted within the same area of Italy [9,10,11,12,13,14,15,16,17,18,19,20,21,22,23,24,25,26,27,28,29,30] with the exception of two studies that were conducted in France [31] and Sweden [32]. An important reason for validating the MPI in different geographical settings is due to possible variations in admission criteria and thresholds under different health care settings. This may result in in-hospital patient populations that are substantially different, particularly regarding co-morbidity burden and frailty, from those investigated in the original studies. 

Additional potential limitations are related to two specific MPI domains, medication use and cognitive function. The medications domain of the MPI collects information regarding the total number of medications a patient is currently taking [9]. The concomitant use of ≥5 medications, known as polypharmacy, has been shown to independently predict adverse outcomes, such as falls and mortality in older adults [33,34]. However, other studies have failed to show significant associations between the total number of medications and the previously mentioned adverse outcomes [35,36,37,38]. Assessing exposure to specific drugs known to increase the risk of physical and/or cognitive decline, such as anticholinergics, with validated scoring systems could identify an area of the MPI to potentially further improve the predictive accuracy [39]. The assessment of cognitive function in the MPI is based on the short portable mental state questionnaire (SPMSQ); however, using this instrument might be problematic in a culturally diverse patient population as the SPMSQ relies on known cultural knowledge. Using a cognitive assessment tool that is not influenced by language, gender or educational status such as the Rowland University Dementia Assessment Scale (RUDAS) might allow a more accurate assessment of cognitive function and, therefore, better predictive capacity [40]. 

As no information is currently available about the applicability, predictive accuracy and validity of the MPI in Australian cohorts we conducted a validation study of the original instrument, and of adapted versions that accounted for the use of specific drug classes and cultural diversity in the assessment of cognitive function, in a population of older in-patients. This information is essential to determine whether this instrument could be easily and accurately applied to Australian patient populations or whether additional modifications are warranted.

## 2. Methods

### 2.1. Study Population

This was a prospective cohort study conducted according to the Declaration of Helsinki and guidelines for Good Clinical Practice. Approval for the study was obtained from local Ethics Committee (reference number 170.15). The study was conducted at Flinders Medical Centre (FMC), a 593-bed metropolitan teaching and trauma hospital within the Southern Adelaide Local Health Network that has a catchment area of approximately 350,000 people. All patients aged ≥ 65 years who presented to Flinders Emergency Department, were admitted to FMC Acute Medical Unit (AMU) and then transferred under either General Medical or Acute Care of the Elderly (ACE) wards from 14 September 2015 to 17 February 2017, were screened for eligibility. ACE wards provide a comprehensive individualised approach in assessing older frail medical in-patients using a multidisciplinary team involving physiotherapists, occupational therapists, social workers, nutrition and dietetics, nursing staff, and geriatricians. Inclusion criteria were: (i) ability to provide informed consent or ability of a proxy for informed consent; and (ii) no diagnosis of dementia prior to hospital admission. 

### 2.2. Original MPI

The 63-item MPI, a prognostic tool based on a standard CGA, was assessed in all study participants in General Medical or ACE wards within the first three days of hospital admission [9]. The MPI score, ranging from 0 to 1, is based on averaging the standardised scores obtained from the eight core domains of the CGA. These eight domains consist of co-habitation status (living alone, with family/friends or in an institute); the total number of medications (taken at admission); functional status evaluated with activities of daily living (ADL) [6] and the instrumental ADL (IADL) [41] scales; cognitive status evaluated by the Short Portable Mental Status Questionnaire (SPMSQ) [42]; evaluation of pressure sores using the Exton Smith Scale (ESS) [43]; co-morbidities assessed using the Cumulative Illness Rating Scale (CIRS) [44]; and nutritional status evaluated by the Mini Nutritional Assessment (MNA) [45]. Each of the domains were given conventional cut-off points derived from the literature with points then allocated as 0 = no problems, 0.5 = minor problems and 1 = major problems. The continuous MPI score (based on the sum of the eight domains’ points and then divided by eight) was then categorised into three risk groups (mild = 0.0–0.33; moderate = 0.34–0.66; severe = 0.66–1.0) [9]. A printable version was used for bedside interviews, with minor adaptations to the SPMSQ domain where ‘President/Pope’ was changed to ‘Prime Minister’. 

### 2.3. Adapted MPI

#### 2.3.1. Medications with Anticholinergic Effects

In addition to assessing the total number of medications, medication use was further characterised by identifying specific drugs with anticholinergic effects, using the validated exposure scoring system Anticholinergic Risk Scale (ARS) [39]. The ARS ranks the anticholinergic effect of each drug on a scale of 0 (limited or none), 1 (moderate), 2 (strong), and 3 (very strong), based on the dissociation constant for the muscarinic receptor, rates of anticholinergic effects vs. placebo in experimental studies and a literature review on anticholinergic adverse effects. The ARS score is calculated by summing the ARS rankings assigned for each of the prescribed drugs in a patient [9]. 

#### 2.3.2. RUDAS

The six-item RUDAS was used as an additional assessment for cognitive function with a score out of 30 points [46]. Due to the array of cultural, religious, and language backgrounds in the Australian population this tool has been specifically designed to minimise these effects and gain a more accurate assessment of cognitive function. It takes approximately 10 minutes to complete.

#### 2.3.3. Adapted MPI Models

Three models were assessed:MPI with number of medications domain substituted with the ARS score. Cut-off points were applied to the ARS score with 0 points for patients not on any anticholinergics, 0.5 points for patients on 1–2 anticholinergics, and 1 point for patients taking >2 anticholinergics.MPI with SPMSQ domain substituted with the RUDAS score. Cut-off points were applied to the RUDAS score with 0 points for patients who had a RUDAS score >25 points, 0.5 points for patients who had a RUDAS score between 25 and 17 points, and 1 point for patients who had a RUDAS score < 17 points.MPI with both the number of medications and SPMSQ domains substituted with ARS and RUDAS scores using the above cut-off points.

### 2.4. Outcome Measures

The primary outcome was all-cause mortality within 12-months. All-cause mortality was defined as death from any cause. This was obtained using the Australian national death registry. This information was collected and updated every three months. 

A number of secondary outcomes were included: in-hospital all-cause mortality, in-hospital falls, and delirium. In-hospital falls were defined as any fall reported in the medical notes. Information regarding the number of falls during the assessment admission was collected together with time elapsed (in days) until first fall and consequences of the fall (none, soft tissue damage, fracture). Delirium was identified as present if the medical team stated this diagnosis in the medical notes. The presence of delirium at admission was recorded to determine if delirium was present pre-admission or developed during admission. Any uncertainties were discussed with a qualified clinician (A. A. M.). This method has been recently highlighted as a more reliable way of capturing the incidence of delirium as opposed to using IDC coding [47]. Other secondary outcomes were length of hospital stay (in days) and hospital re-admission rates for one-, three-, and six-months.

### 2.5. Statistical Analysis

Based on Pilotto’s derivation study, a sample size of *N* = 750 patients was sufficient to provide 80% statistical power to detect an odds ratio of 1.55 for each category increase in the original MPI using a two-sided Type 1 error rate of *p* < 0.05, assuming a 12-month mortality rate of 5.7% in the lowest MPI category [9]. 

Patients’ baseline characteristics were reported as mean ± standard deviations (SD), or frequency and percentages. Comparison of baseline values of the original MPI, RUDAS, ARS, and AMT for subjects that completed follow-up and those that were lost to follow-up were assessed using a Mann–Whitney U test. Differences in in-hospital mortality, gender and mortality at 1-, 3-, 6-, and 12-months were compared using chi-squared tests (see Appendix A). A Kruskall–Wallis test was used to compare age and length of hospital stay (LOS, in days) across MPI risk categories. In-hospital mortality, mortality at 1-, 3-, 6-, and 12-months, falls, and the incidence of delirium was compared across the MPI risk categories using chi-squared tests of independence. 

Simple and multiple logistic regression with adjustment for age and sex was conducted to assess the association between the original MPI and 12-month all-cause mortality. We adjusted for these variables as neither age nor sex is included in the original MPI. Furthermore, a recent study reported a higher prevalence of women in the severe risk category [48].

Area under the receiver operating characteristic (ROC) curve was used to measure the diagnostic accuracy of the original MPI. In order to assess whether the diagnostic value of the adapted MPI was superior to original MPI, comparison of the resultant area under the ROC curves was conducted. Secondary outcomes (in-hospital falls, in-hospital delirium, in-hospital mortality, one-, three-, six-month all-cause mortality, and one-month re-admission rate) were also assessed using logistic regression and area under ROC curves.

Cox proportional hazards analysis adjusted for age and sex was conducted as a sensitivity analysis to assess the risk of all-cause mortality across the original MPI risk categories whilst accounting for total follow-up time. Follow-up time was defined as the time since hospital discharge (when the original MPI was assessed) until the end of follow-up (30 June 2018) or death, whichever occurred first. Estimated survival was also presented graphically using Kaplan–Meier curves. 

Simple and multiple Poisson regression adjusted for age and sex was conducted to compare incidence of in-hospital falls, and readmissions within three- and six-months across the original MPI risk groups. 

Data analyses were performed using the STATA statistical software Version 15.1 (StataCorp©, College City, Texas, USA).

## 3. Results

### 3.1. Study Patient Characteristics

Between 14 September 2015 and 17 February 2017, 738 medical in-patients received an MPI assessment at FMC. Forty-one patients were excluded from the analysis (*N* = 1 with incomplete original MPI, and *N* = 40 lost to follow-up). The final study cohort included 697 patients (348 men and 349 women) with a median age of 80 (Interquartile range (IQR): 72–86), and an age range between 65 and 102 years. No difference in patient characteristics was observed in the 40 patients lost to follow-up compared to the included patient cohort (*n* = 697) (Appendix A).

Most patients were included in the moderate MPI risk category (57.8%) while a third were in the mild risk category (33.9%) and a relatively small proportion in the severe risk category (8.3%) (Table 1). Higher original MPI scores were significantly associated with older age (*p* < 0.001), female sex (*p* < 0.001), in-hospital mortality (*p* < 0.001), delirium (*p* < 0.001), falls (*p* = 0.021), longer LOS (*p* = 0.001), and higher mortality after one month (*p* < 0.001), 3 months (*p* < 0.001), six months (*p* < 0.001), and 12 months (*p* < 0.001). Re-admission within three months (*p* = 0.022), and six months (*p* = 0.0079) were also significantly different across the MPI risk groups. Re-admission rates for one-month did not significantly differ between MPI risk groups (*p* = 0.1572).

### 3.2. Validation of Original MPI—12-Month All-Cause Mortality

In simple logistic regression analysis, the MPI was associated with 12-month all-cause mortality (mild: OR reference group; moderate: OR 2.50, 95% CI: 1.67–3.75; severe: OR 4.24, 95% CI: 2.28–7.88). Multiple logistic regression adjusted for age and sex were significant for 12-month all-cause mortality (Appendix A). The median (inter-quartile range) of follow-up time was 1.62 (0.74–2.45) years and the hazard ratios for all-cause mortality from the Cox proportional hazards regression analysis were similar, although slightly smaller, to the odds ratios for all-cause mortality (Appendix A and Figure 1). 

The diagnostic accuracy (area under the ROC curve) for the MPI adjusted for age and sex was 0.639 (95% CI: 0.593–0.684; Figure 1).

### 3.3. Adapted MPI 

Substituting the total number of medications and the SPMSQ domain with the ARS/RUDAS scores, respectively, into the original MPI, also resulted in an association with 12-month all-cause mortality (Appendix A). There were no differences when comparing the original MPI to the adapted MPI’s area under the ROC curve (original MPI/adapted MPI with ARS substitution *p* = 0.619; original MPI/adapted MPI with RUDAS substitution *p* = 0.958; original MPI/adapted MPI with ARS and RUDAS substitution *p* = 0.673). Adjusting for age and sex did not substantially change the area under the ROC curves (Appendix A). 

### 3.4. MPI and Secondary Outcomes

The MPI was associated with one-, three-, and six-month all-cause mortality (Table 2). The area under the ROC curve for the MPI and one-, three-, and six-month all-cause mortality was 0.708 (95% CI: 0.63–0.79), 0.671 (95% CI: 0.61–0.73), and 0.621 (95% CI: 0.58–0.66), respectively.

In simple logistic regression, the MPI was associated with in-hospital all-cause mortality, but not with in-hospital delirium and one-month re-admission rate (Table 2). The area under the ROC curve for the MPI and in-hospital all-cause mortality and in-hospital delirium was 0.719 (95% CI: 0.64–0.80) and 0.634 (95% CI: 0.57–0.70), respectively.

A total of 95 (13.6%) patients were re-admitted within one-month of which 77 (11.0%) had a single re-admission and 18 (2.6%) had multiple re-admissions. The top three most common reasons for readmission within one month were congestive heart failure (CHF; 7.4%), non-ST elevation myocardial infarction (NSTEMI; 3.2%), hypotension (3.2%), pneumonia unspecified (3.2%), chronic obstructive pulmonary disease (COPD) with acute lower respiratory infection (3.2%), acute kidney failure (3.2%), and neoplastic (malignant) related fatigue (3.2%). A total of 206 (29.56%) patients were re-admitted within three months, of which 133 (19.08%) had a single re-admission and 73 (10.47%) had multiple re-admissions. The top three most common reasons for readmission within three months were CHF (10.2%), COPD with acute lower respiratory infection (10.2%), and pneumonia unspecified (5.8%). A total of 291 (41.75%) patients were re-admitted within six months, of which 159 (22.81%) had a single re-admission and 132 (18.93%) had multiple re-admissions. The top three most common reasons for readmission within 6 months were COPD with acute lower respiratory infection (13.4%), CHF (10%), and pneumonia unspecified (8.2%). The IRR for the original MPI for re-admissions within 1 month was not significant for any risk group (Table 2). The IRR for the original MPI for re-admissions within there or six months was significant for the moderate risk group compared to the mild risk group but not compared to the severe risk group (Table 2). 

Amongst the 20 (2.9%) patients who had a fall in-hospital, 19 (2.73%) were single fallers and one1 (0.14%) was a recurrent faller. The IRR for the MPI was not associated with number of in-hospital falls (Table 2).

## 4. Discussion

In this study of medical in-patients aged ≥65 years we demonstrated that the original MPI in a geographically different patient population showed inferior discrimination to that of the original derivation and validation study for 12-month mortality [9]. A relatively low discrimination was also observed with secondary outcomes (one-, three-, six-month all-cause mortality, delirium, and number of re-admissions within one month). However, in-hospital all-cause mortality showed similar discrimination to that observed in the original MPI derivation study [9]. Adaptations to the MPI with SPMSQ and number of medications domains substituted with ARS and RUDAS scores did not significantly change discrimination for 12-month all-cause mortality. 

For our FMC hospital cohort, the predictive performance of 12-month all-cause mortality for the MPI was relatively lower compared to other validation studies [11,16,18,24,25]. These studies were similar in design to the original MPI derivation study [9]. However, caution is required when interpreting the results of those studies as three studies had a small sample size, between 134–262 patients [11,24,25], and all studies were conducted in patient populations with specific medical conditions (CAP [11], dementia [25], TIA [18], and oncology patients [24] with higher overall mortality risk. Although most of the clinical and demographic characteristics in our cohort were similar to those in Pilotto’s cohort [9], some differences were also identified. For example, in our study cohort a greater proportion of patients were allocated into the moderate risk group than in Pilotto’s cohort. Furthermore, for low and moderate risk patients in the FMC cohort, higher mortality rates were seen for six- and 12-month all-cause mortality when compared to Pilotto’s cohort. Additionally, the original MPI in our study was characterized by higher scores of the CIRS components and a greater number of medications when compared to Pilotto’s cohort. 

An arising concern in geriatric patients is the increasing evidence that certain medications, in particular anticholinergics, with specific pharmacological effects might more strongly predict physical and cognitive decline and mortality in older adults than the total number of drugs [49,50]. Therefore, the MPI was also adapted in this study by substituting the number of drugs with the ARS score [39]. In addition, the cognitive screen test (SPMSQ) in the MPI was substituted with the RUDAS which is a frontal lobe assessment tool, suitable for a multicultural population [46]. Although the adapted MPI was associated with 12-month mortality, the diagnostic accuracy was not significantly different to that of the original MPI. 

The original MPI was also associated with one- , three-, and six-month all-cause mortality and the discrimination for six-month all-cause mortality was similar to that for 12-month all-cause mortality. Discrimination for one- and three-month all-cause mortality was slightly higher and discrimination for one-month mortality was similar to the original MPI for similar patient population studies [10,11,18,25]. Again, these other studies were small sample size (ranging from 134 to 654 patients) and all studies were conducted in patient populations with specific medical conditions (GI bleed [10], CAP [11], dementia [25], and TIA patients [18]).

The original MPI was also associated with in-hospital all-cause mortality and the discrimination was similar to other studies [21,26]. At present, no study has assessed the performance of the original MPI in predicting in-hospital falls and delirium and we observed no association within this study. However, the study was not specifically powered for these two outcomes and there were only a relatively small number of these events that occurred.

There was a significant association between the original MPI and LOS with patients in the higher MPI categories having an increased length of stay. The mean LOS for our cohort was shorter within each MPI category when compared to other validation studies [21,26]. These findings agree with that of another study in which the original MPI performance for LOS was similar to that for 12-month all-cause mortality when LOS was treated as a binary outcome (≤10 days versus >10 days). However, the study excluded patients who died within hospital [26]. 

At present, there are no studies that have assessed the performance of the MPI for predicting re-admission rate for one-, three- and six-months. Within our study, only the original MPI moderate risk category was associated with the three- and six-month re-admission rates. However, our study was not specifically powered for either of these secondary outcomes.

This study has several limitations. Not all patient outcomes may have been captured since the data was obtained from a South Australia specific database and, therefore, patients that moved outside of South Australia (SA) or were admitted to a private hospital within SA or interstate may have been missed, resulting in an underestimation of deaths and potential bias in our estimates. In addition, our findings represent current practices at a single centre and may not be a true representation of other Australian populations. In particular, thresholds for FMC hospital admission could differ from the derivation study [9] with FMC potentially only admitting more severely ill patients, resulting in a greater number of patients being assigned into the moderate risk group of the MPI. Another limitation was that no information regarding ethnicity and racial differences was specifically captured in our study population. 

Nevertheless, this study has its strengths. Firstly, this is a large cohort outside of Europe that provided well described data and allowed for adjustment for important confounders. Secondly, there were a number of sensitivity analyses performed with different follow-up time-points and adapted versions of the original MPI to analyse its purpose in an Australian population.

## 5. Conclusions

In conclusion, our study raises some concerns regarding the predictive performance of the MPI for use in an Australian population that included a heterogeneous population of older patients. In particular, the discriminative performance of the MPI for 12-month mortality was lower than that reported in the original MPI studies. Adaptation of the MPI by substituting the number of medications and the SPMSQ with the ARS and RUDAS scores did not substantially improve the performance from the original MPI. An exploratory analysis of in-hospital falls, delirium, and re-admissions suggest that the original MPI might also be of value in assessing these outcomes within future studies. Larger prospective multi-centre studies with similar patient populations are now required to confirm our findings before implementing the use of the MPI into routine clinical practice within Australia.

## Figures and Tables

**Figure 1 jcm-08-01820-f001:**
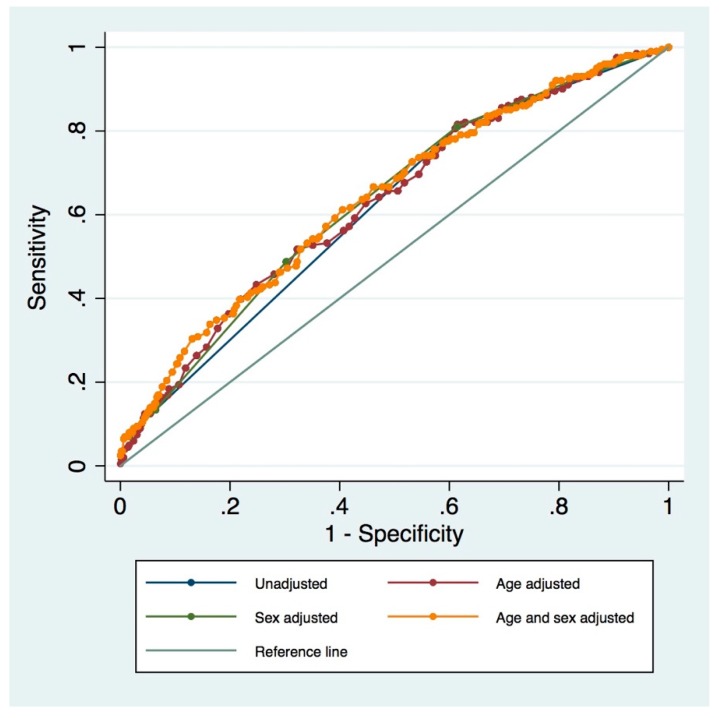
Unadjusted and adjusted ROC curves for original MPI and 12-month all-cause mortality.

**Table 1 jcm-08-01820-t001:** Patient characteristics according to MPI category.

	Mild Risk	Moderate Risk	Severe Risk	*p*-Value
Characteristics	0.0–0.33	0.34–0.66	0.67–1.0	
Patients, n (%)	229 (32.86)	409 (58.68)	59 (8.46)	
Women	92 (40.17)	220 (53.79)	37 (62.71)	<0.001
Men	137 (59.83)	189 (46.21)	22 (37.29)	
MPI score				
Score range	0.063–0.313	0.375–0.625	0.688–0.875	
Median (IQR)	0.250 (0.250–0.313)	0.438 (0.438–0.500)	0.688 (0.688–0.750)	
Age				
Range	65–96	65–101	67–102	
Median (IQR)	75 (70–82)	82 (74–87)	86 (80–92)	<0.001
Fall, *n* (%)	4 (1.75)	11 (2.69)	5 (8.47)	0.021
Delirium, *n* (%)	14 (6.11)	34 (8.31)	19 (32.20)	<0.001
LOS (days), median (IQR)	5 (3–8)	6 (4–11)	11 (5–18)	0.001
Mortality, *n* (%)				
In-hospital	1 (0.44)	16 (3.91)	8 (13.56)	<0.001
1-month	3 (1.31)	20 (4.89)	12 (20.34)	<0.001
3-month	7 (3.06)	42 (10.27)	16 (27.12)	<0.001
6-month	22 (9.61)	95 (23.23)	20 (33.90)	<0.001
12-month	38 (16.59)	136 (33.25)	27 (45.76)	<0.001
Re-admission rate, %				
1-month	11.35	15.65	8.47	0.157
3-month	23.58	33.25	27.12	0.022
6-month	35.37	45.72	38.98	0.008

Abbreviations: IQR: Interquartile range; LOS: length of stay; n: number. Kruskal–Wallis H test: age, gender, and LOS; Chi2 test: in-hospital mortality, 1-, 3-, 6-, and 12-month mortality, falls, delirium, re-admissions at 1, 3 and 6 months.

**Table 2 jcm-08-01820-t002:** FMC cohort and secondary outcomes for MPI risk categories.

	Unadjusted OR			Adjusted OR *		
**Variable**						
Six-month mortality						
Mild	Reference group					
Moderate	2.85	1.73–4.67	<0.0001	2.92	1.75–4.87	<0.0001
Severe	4.83	2.41–9.67	<0.0001	4.88	2.34–10.20	<0.0001
Three-month mortality						
Mild	Reference group					
Moderate	3.63	1.60–8.22	0.002	3.56	1.55–8.20	0.003
Severe	11.80	4.58–30.40	<0.0001	11.20	4.14–30.32	<0.0001
One-month mortality						
Mild	Reference group					
Moderate	3.87	1.14–13.18	0.030	3.91	1.13–13.55	0.031
Severe	19.23	5.22–70.83	<0.0001	19.43	4.96–76.06	<0.0001
In-hospital mortality						
Mild	Reference group					
Moderate	9.28	1.22–70.46	0.031	9.19	1.19–70.74	0.033
Severe	35.76	4.38–292.33	0.001	34.46	4.02–295.78	0.001
In-hospital delirium						
Mild	Reference group					
Moderate	1.39	0.73–2.65	0.314	1.21	0.62–2.37	0.573
Severe	7.29	3.38–15.73	<0.0001	5.68	2.48–13.01	<0.0001
Unadjusted IRR				Adjusted IRR *		
	IRR	95% CI	*p*-value	IRR	95% CI	*p*-value
Re-admission rate, one month						
Mild	Reference group					
Moderate	1.29	0.86–1.94	0.223	1.31	0.86–2.00	0.215
Severe	0.59	0.23–1.51	0.269	0.60	0.23–1.58	0.304
Re-admission rate, three months						
Mild	Reference group					
Moderate	1.46	1.14–1.88	0.003	1.52	1.17–1.97	0.002
Severe	0.88	0.53–1.44	0.608	0.95	0.57–1.59	0.844
Re-admission rate, six months						
Mild	Reference group					
Moderate	1.59	1.31–1.93	<0.0001	1.69	1.39–2.07	<0.0001
Severe	1.04	0.72–1.50	0.830	1.18	0.81–1.72	0.379
In-hospital falls						
Mild	Reference group					
Moderate	1.24	0.40–3.85	0.706	1.04	0.32–3.33	0.949
Severe	2.24	0.56–8.95	0.254	1.69	0.40–7.18	0.478

Abbreviations: IRR: Incidence rate ratio; CI: confidence interval; MPI: Multidimensional Prognostic Index; OR: Odds ratio. Note: * adjusted for age and gender.

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
