# Peer review of "Validation and Adaptation of the Multidimensional Prognostic Index in an Older Australian Cohort"

_jcm, 2019, doi:10.3390/jcm8111820_

Round 1

Reviewer 1 Report

The authors have conducted a validation study of the original Multidimensional Prognostic Index (MPI), which is an objective and quantifiable tool based on the Comprehensive Geriatric Assessment, and of adapted versions, in a population of older inpatients, in Australia. They showed that the original and the adapted versions of MPI were strongly associated with 12-month mortality (and with other important outcomes) in this elderly cohort; however, the discriminatory performance was lower than reported in the original studies conducted in Europe.

Please find below a few minor comments:

(1) The authors have used the terms “univariate and multivariate” logistic regression analysis. The correct term in this occasion is “multivariable”.

Multivariate analysis involves simultaneous analysis of more than one outcome variable. On the other hand, multivariable analysis assesses the relationship between one dependent (outcome) variable and several independent variables.

To avoid confusion, I would suggest using the terms “simple” and “multiple” logistic regression, throughout the manuscript.

(2) In page 4, lines 181-182, the authors write that “The final study cohort included 697 patients (367 men and 370 women)”. Please correct the numbers of men and women.

(3) Table 2, top. There is some problem, I guess, in the first part of the table referring to LOS (days). Infact, the outcome variable must be “discharge from hospital”, not “LOS (days)”. That’s why you have analyzed with Cox regression and report Hazard Ratios.

If so, what is the reference group (with hazard = 1.0).

Why you give a HR for each one of the 3 MPI risk categories? Please clarify.

Also, include HR in the respective abbreviations list (bottom of Table 2).

Moreover, please use the terms “upper and lower confidence limit” instead of “upper and lower confidence interval”.

Reviewer 2 Report

Interesting article. However, I need the following questions to be addressed for the relevance of this article.

Major points:

You are stating in the method section that you are using ARS and RUDAS in place of two of the domains of MPI. If the aim of this study is to see the relationship between medication use/cognitive function and mortality/LOS/number of falls/etc., why not simply compare ARS/RUDAS with these outcomes (instead of using all eight categories of MPI) or adjust the 6 categories of MPI and only focus on ARS and RUDAS as independent variables? Otherwise, if the aim of this study is to compare MPI and the outcomes stated in the text, what is the point of substituting ARS/RUDAS for MPI calculation? age range is too wide (ex. a patient at age of 65 has more chance of surviving than a patient of age 102). We can intuitively guess that MPI and age are related. So, why not stratify according to age categories and calculate odds ratio in each age group? likewise, can we disregard the effect of age and simply compare the length of hospital stay according to the severity of MPI? What is the interpretation for the age difference (% of women) in each MPI category? (why are there more women in the severe risk category?) Regarding the ROC to see the accuracy of MPI in predicting the 12-month mortality, do you mean that anyone who dies within 12 months is counted as "dead" and others as "survived"? If so, is it relevant to see the outcome (death) the same regardless of the timing of death? (is a patient who dies at 1 month of follow up the same as one who dies at 12 month of follow up?) If (as you state in the introduction) the new point of this research is that this is the first non-European study of the subject, you should consider the ethnic/racial differences within the patients enrolled in the study.

Minor points:

In the last paragraph of introduction part (lines 71-74), you should state what you are going to do in your research (rather than the fact from the literature). What are the reasons for re-admissions? All-cause? or something related to injuries? the statistical analysis can be sub-divided into several sections according to what you want to measure (eg. mortality, LOS, etc).

Round 2

Reviewer 2 Report

As you responded to my comment #2, the higher prevalence of women in the severe risk category has also been observed in a recent paper1. You need to state that in the method section (with the reference to Veronese et.al.).

Author Response

Comment:

As you responded to my comment #2, the higher prevalence of women in the severe risk category has also been observed in a recent paper1. You need to state that in the method section (with the reference to Veronese et.al.).

Response:

To address this comment we have added two sentences after the following (lines 163-164): Simple and multiple logistic regression with adjustment for age and sex was conducted to assess the association between the original MPI and 12-month all-cause mortality.”

Changes:

We have added the following sentences and citation in the methods section (line 164): “We adjusted for these variables as neither age or sex is included in the original MPI. Furthermore, a recent study reported a higher prevalence of women in the severe risk category [48].”